# Unsupervised Clustering using Pseudo-semi-supervised Learning

**Divam Gupta** [*]
Carnegie Mellon University
divam@cmu.edu

**Ramachandran Ramjee**
Microsoft Research India
ramjee@microsoft.com

**Nipun Kwatra**
Microsoft Research India
nipun.kwatra@microsoft.com

**Muthian Sivathanu**
Microsoft Research India
muthian@microsoft.com

## Abstract

In this paper, we propose a framework that leverages semi-supervised models to improve unsupervised clustering performance. To leverage semi-supervised models, we first need to automatically generate labels, called pseudo-labels. We find that prior approaches for generating pseudo-labels hurt clustering performance because of their low accuracy. Instead, we use an ensemble of deep networks to construct a similarity graph, from which we extract high accuracy pseudo-labels. The approach of finding high quality pseudo-labels using ensembles and training the semi-supervised model is iterated, yielding continued improvement. We show that our approach outperforms state of the art clustering results for multiple image and text datasets. For example, we achieve 54.6% accuracy for CIFAR-10 and 43.9% for 20news, outperforming state of the art by 8-12% in absolute terms. Project details and code are available at https://divamgupta.com/pseudo-semi-supervised-clustering

## 1 Introduction

Semi-supervised methods, which make use of large unlabelled data sets and a small labelled data set, have seen recent success, e.g., ladder networks Rasmus et al. (2015) achieves 99% accuracy in MNIST using only 100 labelled samples. These approaches leverage the unlabelled data to help the network learn an underlying representation, while the labelled data guides the network towards separating the classes.

In this paper, we ask two questions: *is it possible to create the small labelled data set required by semi-supervised methods purely using unsupervised techniques? If so, can semi-supervised methods leverage this autonomously generated pseudo-labelled data set to deliver higher performance than state-of-the-art unsupervised approaches?* We answer both these questions in the affirmative.

We first find that prior approaches for identifying pseudo-labels Caron et al. (2018); Chen (2018); Lee (2013) perform poorly because of their low accuracy (Section 2). To create a high accuracy pseudo-labelled data set autonomously, we use a combination of ensemble of deep networks with a custom graph clustering algorithm (Section 4). We first train an ensemble of deep networks in an unsupervised manner. Each network independently clusters the input. We then compare two input data points. If all of the networks agree that these two data points belong to the same cluster, we can be reasonably sure that these data points belong to the same class. In this way, we identify all input data pairs belonging to the same class with high precision in a completely unsupervised manner.

In the next step, we use these high quality input pairs to generate a similarity graph, with the data points as nodes and edges between data points which are deemed to be similar by our ensemble. From this graph, we extract tight clusters of data points, which serve as pseudo-labels. Note that, in this step, we do not cluster the entire dataset, but only *a small subset on which we can get high*

---

[*]Work done as a Research Fellow at Microsoft Research India

*precision*. Extracting high quality clusters from this graph while ensuring that the extracted clusters correspond to different classes is challenging. We discuss our approach in Section 4.2.1 for solving this problem. In this way, our method extracts unambiguous samples belonging to each class, which serves as pseudo-labels for semi-supervised learning.

For semi-supervised learning using the labels generated above, one could use ladder networks Rasmus et al. (2015). However, we found that ladder networks is unsuitable for the initial unsupervised clustering step as it can degenerate to outputting constant values for all inputs in the absence of unsupervised loss. To enable unsupervised clustering, we augment ladder networks using information maximization Krause et al. (2010) to create the *Ladder-IM*, and with a dot product loss to create *Ladder-Dot*. We show in Section 5 that *Ladder-IM* and *Ladder-Dot*, by themselves, also provide improvements over previous state of the art. We use the same models for both the first unsupervised learning step as well as the subsequent pseudo-semi-supervised iterations.

Finally, the approach of finding high quality clusters using an ensemble, and using them as labels to train a new ensemble of semi-supervised models, is iterated, yielding continued improvements. The large gains of our method mainly come from this iterative approach, which can in some cases, yield upto 17% gains in accuracy over the base unsupervised models (see section 5.4). We name our pseudo-semi-supervised learning approach *Kingdra*[1]. *Kingdra* is independent of the type of data set; we show examples of its use on both image and text data sets in Section 5. This is in contrast to some previous approaches using CNNs, e.g. Chang et al. (2017), Caron et al. (2018), which are specialized for image data sets.

We perform unsupervised classification using *Kingdra* on several standard image (MNIST, CIFAR10, STL) and text (reuters, 20news) datasets. *On all these datasets, Kingdra is able to achieve higher clustering accuracy compared to current state-of-the-art deep unsupervised clustering techniques.* For example, on the CIFAR10 and 20news datasets, *Kingdra* is able to achieve classification accuracy of 54.6% and 43.9%, respectively, delivering 8-12% absolute gains over state of the art results Hu et al. (2017); Xie et al. (2016).

## 2    PRIOR WORK ON GENERATING PSEUDO-LABELS

| Model + Pseudo-label Method | MNIST | | CIFAR10 | |
|---|---|---|---|---|
| | Label Acc.(%) | Cluster Acc. (%) | Label Acc. (%) | Cluster Acc. (%) |
| *Ladder-IM* + Argmax (Lee (2013)) | 95.4 | 95.4 | 49.0 | 49.0 |
| *Ladder-IM* + K-means (Caron et al. (2018)) | 75.4 | 60.9 | 45.3 | 44.8 |
| *Ladder-IM* + Threshold (Chen (2018)) | 88.6 | 91.6 | 60.5 | 47.4 |

Table 1: Pseudo-label and clustering accuracy

Several techniques have been proposed in the literature for generating pseudo-labels (Caron et al. (2018); Chen (2018); Lee (2013). In Lee (2013), the output class with the highest softmax value (Argmax) is taken to be the pseudo-label. In Caron et al. (2018), the authors perform K-means clustering on the feature vector and use the K-means clusters as pseudo-labels. Finally, authors in Chen (2018) treat the softmax output as confidence and only label those items whose confidence value is above a high threshold. Note that none of these techniques for identifying pseudo-labels have been applied in our context, i.e., for unsupervised clustering using semi-supervised models.

In this section, we evaluate if pseudo-labels created by these prior techniques can be leveraged by semi-supervised models to improve clustering accuracy. We start with a semi-supervised model based on Ladder networks (Rasmus et al. (2015)) called *Ladder-IM* (see Section 4.1 for model details) and train using only its unsupervised loss terms on MNIST and CIFAR10 datasets. We use each of the above three pseudo-labelling approaches on the trained model to provide an initial set of pseudo-labels to the datasets (e.g., using K-means clustering on the feature vector of the model as in Caron et al. (2018), etc.). We call the accuracy of these pseudo-labels the initial pseudo-label accuracy. We then use these generated pseudo-labels along with the datasets to train the model again,

---

[1]Our system is named after a semi-pseudo Pokémon.

now with a supervised loss term (based on the pseudo-labels) and the earlier unsupervised loss terms. We again run the pseudo-labelling approaches on the newly trained model to derive an updated set of pseudo-labels. We iterate this process of training and pseudo-labelling until the pseudo-label accuracy stabilizes. We call this the final clustering accuracy.

The initial pseudo-label accuracy and the final clustering accuracy results for the three approaches are shown in Table 1. First, consider MNIST. The unsupervised clustering accuracy of *Ladder-IM* is 95.4%. Argmax simply assigns pseudo-labels based on the model's output and since this doesn't add any new information for subsequent iterations, the final accuracy remains at 95.4%. On the other hand, the pseudo-labels identified by both the K-means and threshold approaches result in worse initial label accuracy (75.4% and 88.6%). When this low-accuracy pseudo-label is used as supervision to train the model further, it results in a low final clustering accuracy of 60.9% and 91.6%, respectively. CIFAR10 results are similar. *Ladder-IM* clustering accuracy is 49% which remains the same under Argmax as before. Pseudo-label accuracy using the K-means approach is worse and results in pulling down the final accuracy to 44.8%. Interestingly, threshold results in a slightly higher initial accuracy of 60.5% but even this is not high enough to improve the final clustering accuracy for CIFAR10.

From these results, we arrive at the following two conclusions. First, if the initial pseudo-label accuracy is not high, using pseudo-labels as supervision can result in bringing down the final clustering accuracy. Thus, high accuracy of initial pseudo-labels is crucial for improving clustering accuracy. Second, current approaches for identifying pseudo-labels do not deliver high accuracy and hence are unable to help improve overall clustering accuracy.

## 3 RELATED WORK

**Unsupervised clustering:** Various unsupervised clustering methods have been proposed over the years. Ng et al. (2002) uses a spectral clustering based approach, while Elhamifar & Vidal (2009) uses a sparse subspace approach for unsupervised learning. Recently, several deep neural networks based methods have been proposed, which scale well to large datasets. The ability of deep neural networks to learn higher level representations make them a good choice for unsupervised learning. Coates & Ng (2012) and Caron et al. (2018) use convnets and k-means for clustering. Caron et al. (2018) for example, iterates over clustering the features obtained from a convnet, and training the classifier using these clusters as pseudo-labels. The authors do not report clustering perfomence and we observed that this method can easily degenerate. Chang et al. (2017) uses pair-wise dot-product based similarity to identify close input pairs, which provide a supervisory signal. These convnets based approaches however work on only image datasets. Xie et al. (2016) simultaneously learns feature representations and cluster assignments using deep neural networks, and works on both image and text datasets. Hu et al. (2017) uses regularization combined with mutual information loss for unsupervised learning and achieves state of the art results. The authors conduct experiments in two settings - Random Perturbation Training and Virtual Adversarial Training. Other works such as Hjelm et al. (2018) using mutual information, maximize the mutual information between the spatial features and the non-spatial features. Ji et al. (2019) maximizes the mutual information between the predicted label of the image and the predicted label of the augmented image. This method uses convolution networks and requires domain knowledge of the dataset.

**Self-supervised learning:** Another form of unsupervised learning uses auxiliary learning tasks for which labels can be *self* generated to generate useful representations from data. Many methods use spatial information of image patches to generate self-supervised data. E.g. Pathak et al. (2016) predicts pixels in an image patch using surrounding patches, while Doersch et al. (2015) predicts the relative position of image patches. Sermanet et al. (2018) uses time as a self supervisory signal between videos taken from different view points. Temporal signal is also used to learn representations from single videos by predicting future frames, e.g. Denton et al. (2017). Our method uses correlation between outputs of input points across an ensemble as a supervisory signal to generate self-supervised pseudo-labels.

**Semi-Supervised learning:** Semi-supervised approaches use sparse labelling of datapoints. Szummer & Jaakkola (2002) propagates labels based on nearest neighbors. Weston et al. (2012) uses a deep version of label propagation. Lee (2013) adjusts labels probabilities, starting with trusting only true labels and gradually increases the weight of pseudo labels. Rasmus et al. (2015) employs a denoising

auto encoder architecture and have shown impressive performance. Tarvainen & Valpola (2017) uses an averaged model over previous iterations as a teacher. Other than these, some semi-supervised learning methods like Xie et al. (2019) and Berthelot et al. (2019) use data augmentation and assume some domain knowledge of the dataset with some of the data augmentation specific to image datasets. Miyato et al. (2018) and Shinoda et al. (2017) uses virtual adversarial training combined with the classification loss to perform semi-supervised classification. However, we found that these methods do not work well if we jointly train them with unsupervised losses. Ladder networks does not require any domain-dependent augmentation, works for both image and text datasets, and can be easily jointly trained with supervised and unsupervised losses. Thus, we chose to work with Ladder networks in our experiments, though our approach is general enough to work with any semi-supervised method that accommodates training with unsupervised loss terms.

**Unsupervised ensemble learning:** Unsupervised ensemble learning has been mostly limited to generating a set of clusterings and combining them into a final clustering. Huang et al. (2016) cast ensemble clustering into a binary linear programming problem. Wang et al. (2009); Fred & Jain (2005) use a pair wise co-occurrence approach to construct a co-association matrix and use it to measure similarity between data points. See Vega-Pons & Ruiz-Shulcloper (2011) for a survey of ensemble clustering algorithms. Note that to the best of our knowledge none of the ensemble clustering algorithms employ a semi-supervised step like ours, or make use of deep networks.

## 4 PROPOSED FRAMEWORK

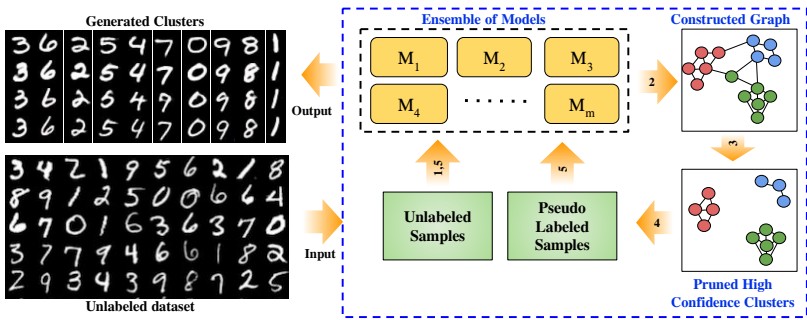

Figure 1: *Kingdra* overview. In step 1, we train all the models using the unlabeled samples. In step 2 we construct a graph modeling pairwise agreement of the models. In step 3, we get $k$ high confidence clusters by pruning out data-points for which the models do not agree. In step 4 we take the high confidence clusters and generate pseudo labels. In step 5 we train the models using both unlabeled samples and pseudo labeled samples. We iterate step 2 to step 5 and final clusters are generated.

An overview of the *Kingdra* method is summarized in Figure 1. Given an unlabelled dataset $\mathbf{X} = \{x_1, \ldots, x_n\}$, we start with unsupervised training of an ensemble of models $\mathbf{M} = \{M_1, \ldots, M_m\}$. For the individual models, any unsupervised model can be used. However, we propose a novel *Ladder-\** model, in which we build on ladder networks Rasmus et al. (2015) and modify it to support clustering. Next, we use the agreement between the ensemble models on a pair of data points, as a measure of similarity between the data points. This pairwise data is used to construct a similarity graph, from which we extract $k$ tight clusters of data points, which serve as pseudo-labels. Note that, in this step, we do not cluster the entire dataset, but only a small subset on which we can get high precision. This is important for improving the accuracy of our semi-supervised training, as discussed in section 2. These pseudo-labels then serve as training data for semi-supervised training of a new ensemble of *Ladder-\** models. Finally, we perform multiple iterations of the above steps for continued improvement.

### 4.1 BASE MODEL

The first step of our method is unsupervised training of an ensemble of models. Our framework allows using any unsupervised method for this step, and we have experimented with existing approaches, such as IMSAT Hu et al. (2017). The accuracy of this base model directly impacts the accuracy of

our final model, and thus using an accurate base model clearly helps. In that light, we have also developed a novel unsupervised model *Ladder-\**, which outperforms other unsupervised models in most data sets.

Ladder networks Rasmus et al. (2015) have shown great success in semi-supervised setting. However, to the best of our knowledge, the ladder architecture has not been used for unsupervised clustering. One reason perhaps is that ladder networks can degenerate to outputting constant values for all inputs in the absence of a supervised loss term. To circumvent this degeneracy, we add an unsupervised loss to the regular ladder loss terms so that it directs the network to give similar outputs for similar inputs, but overall maximizes the diversity in outputs, so that dissimilar inputs are directed towards dissimilar outputs. We achieve this objective by incorporating one of two losses – the IM loss Krause et al. (2010); Hu et al. (2017) or the dot product loss Chang et al. (2017). We call the two variants *Ladder-IM* and *Ladder-Dot*, respectively.

**IM loss:** The IM loss or the information maximization loss is simply the mutual information between the input $X$ and output $Y$ of the classifier:

$$I(X;Y) = H(Y) - H(Y|X) \tag{1}$$

where $H(.)$ and $H(.|.)$ are the entropy and conditional entropy, respectively. Maximizing the marginal entropy term $H(Y)$, encourages the network to assign disparate classes to the inputs, and thus encourages a uniform distribution over the output classes. On the other hand, minimizing the conditional entropy encourages unambiguous class assignment for a given input.

**Dot product loss:** The dot product loss is defined to be

$$D(X_i, X_j) = Y_i^T Y_j, \text{ if } i \neq j \tag{2}$$

which forces the network outputs for different inputs to be as orthogonal as possible. This has a similar effect to IM loss, encouraging the network to assign disparate classes to the inputs.

Among *Ladder-IM* and *Ladder-Dot*, we found *Ladder-IM* to perform better than *Ladder-Dot* in most cases. However, we did find that *Ladder-Dot* along with *Kingdra* iterations outperforms when the data set has a large imbalance in the number of samples per class. The reason for this is that the dot product loss is agnostic to the number of samples per class, while the marginal entropy term in the IM loss will drive the network towards overfitting a class with more samples, compared to a class with less number of samples. A more detailed presentation of *Ladder-IM* and *Ladder-Dot* can be found in the appendix.

### 4.2 Unsupervised Ensembling

*Kingdra* exploits an ensemble of *Ladder-\** models to further improve the performance of unsupervised learning. Note that, in supervised learning, ensembling is trivial as we can simply average the outputs of the individual models or do voting on them. On the other hand, in unsupervised learning, it is not trivial to do voting, as in the absence of training labels there is no stable class assignment for outputs across different models, and thus we do not have any mapping of class IDs of one model to another. To solve this we propose a simple approach, where we look at pairs of data-points, rather than at individual samples. Two data-points are in the same cluster with a high confidence if majority (or all) of the models in the ensembles put them in same cluster. For example, given an input pair $x, x\prime$, if $M_i(x) = M_i(x\prime)$ for enough models, we can say with high confidence that they belong to the same class. Using this pairwise approach, we propose a graph based method to find small sized, but high precision clusters.

#### 4.2.1 Graph Based Mini-Clustering

We construct a graph $G = \{X, E_{pos}, E_{neg}\}$ with $n$ nodes where each input data-point $x$ is represented as a node. Here $E_{pos}$ and $E_{neg}$ are two types of edges in the graph :

- **Strong Positive Edges:** A strong positive edge is added between two data-points when a large number of models agree on their predicted class. $(x, x\prime) \in E_{pos} \iff n\_agree(x, x\prime) \geq t_{pos}$ where $t_{pos}$ is a chosen threshold, and $n\_agree(x, x\prime) = |\{m : m \in \mathbf{M}, m(x) = m(x\prime)\}|$.

- **Strong Negative edges:** A strong negative edge is added between two data-points when a large number of models disagree on their predicted class. $(x, x\prime) \in E_{neg} \iff n\_disagree(x, x\prime) \geq t_{neg}$, where $t_{neg}$ is a chosen threshold, and $n\_disagree(x, x\prime) = |\{m : m \in \mathbf{M}, m(x) \neq m(x\prime)\}|$.

A strong positive edge between two data points, implies that most models believe they are in the same class, while a strong negative edge between two data points implies that most models believe they should belong to different classes.

---

**Algorithm 1** Get high precision clusters using ensembles

---

1: **procedure** GETCLUSTERS($\mathbf{X}$, $k$ )
2:     $G = \{X, E_{pos}, E_{neg}\}$
3:     **for** $k' \in \{1, 2, \ldots, k\}$ **do**
4:         $x_{max} = \text{argmax}_{x \in \mathbf{X}}\{|(x, x') \in E_{pos}|\}$
5:         $S_{k'} = \{x : (x, x_{max}) \in E_{pos}\} \cup \{x_{max}\}$
6:         **for** $x' \in \mathbf{X}$ **do**
7:             Remove $x'$ from $G$, if $(x', x_{max}) \notin E_{neg}$
8:         **end for**
9:     **end for**
10:    Return $\mathbf{S} = \{S_1, S_2, \ldots, S_k\}$
11: **end procedure**

---

After building the graph, each clique of strong positive edges would be a cluster, where within a clique, data-points belong to the same class with high confidence. Since we add only high confidence edges to the graph, the number of cliques can be much larger than $k$. Hence we need to select $k$ cliques where we would like to maximize the size of each clique, but also require that the cliques are diverse (in order to not select two cliques with data-points belonging to the same class). Hence, within a clique, nodes should be connected by strong positive edges, while across cliques, nodes should be connected by strong negative edges. As finding cliques is not solvable in polynomial time, we use a simple and efficient greedy approximation algorithm, as shown in Algorithm 1.

Rather than finding cliques, we greedily find nodes with the highest number of strong positive edges (line 4). The intuition is that most of the neighbours of that node will also be connected with each other. In the case of Cifar-10, we find that with a threshold of 90%, 81% of nodes are fully connected with each other. If the threshold is 100%, all nodes in a cluster are connected with each other by transitivity. We take the node with highest number of strong positive edges, along with other nodes connected to it by strong positive edges and add them to a cluster (line 5). We then remove all the nodes that do not have a strong negative edge to the chosen node (line 6–7). The intuition here is that these nodes are not diverse enough from the chosen cluster (since some models think that they belong to the same class as the currently chosen node), and thus should not be part of the next set of chosen clusters. By repeating the process $k$ times, we get $k$ diverse clusters, approximately satisfying our requirement.

### 4.3 Iterative Ensemble Training

Once the high precision clusters are identified, we treat these clustered points (points in set $\mathbf{S}$) as pseudo-labels, and solve our unsupervised clustering problem using a semi-supervised method. Although any semi-supervised method can be used, as described in section 4.1 we use the proposed *Ladder-\** method, which we found superior to ladder networks in our experiments.

Instead of training a single semi-supervised model, we train an ensemble of models, and again use them to find high quality clusters. This approach can be iterated, yielding continued improvements. We name this approach *Kingdra*. Algorithm 2 describes the complete *Kingdra* algorithm. First, the individual models are trained using only the unsupervised *Ladder-\** loss (lines 1–4). Then, for each of the iterations, we obtain high precision clusters (line 6), derive pseudo-labels from them (line 8), and then train the models with both the unsupervised and supervised losses (lines 9–10).

We compute the pseudo-labels using the mini-clusters as follows. For a model $M_j \in \mathbf{M}$ and clusters $\mathbf{S}$, we need to find an appropriate mapping of the clusters to the output classes of the model. In

---

**Algorithm 2** *Kingdra*: Iterative Ensemble Training

---
    **Input** : Dataset **X**, Models **M**, num_clusters $k$
    **Output** Cluster Labels
 1: **for** $j \in \{1, 2, \ldots, m\}$ **do**
 2:      Initialize weights of $M_j$
 3:      Update $M_j$ by minimizing $loss^{Ladder\text{-}*}$
 4: **end for**
 5: **for** $it \in \{1, 2, \ldots, n\_iter\}$ **do**
 6:      $S = \text{GetClusters}(\mathbf{X}, k)$
 7:      **for** $j \in \{1, 2, \ldots, m\}$ **do**
 8:          Get pseudo labels for $M_j$
 9:          Update $M_j$ by minimizing:
10:          $loss^{Ladder\text{-}*} + loss^{sup}$                 ▷ Use pseudo labels for $loss^{sup}$
11:      **end for**
12: **end for**
13: Use averaging on the ensemble models **M** to return final clusters

---

particular, for a cluster $S\prime \in \mathbf{S}$, we assign all data-points in $S\prime$ the following label:

$$y_{S\prime}^j = \text{mode}(\{M_j(x') : x' \in S'\}). \tag{3}$$

That is, we map a cluster to the output class to which most data-points in the cluster are mapped. These pseudo-labels are then used for computing the supervised loss of *Ladder-\**. This iterative approach leads to a continuous improvement of clustering quality. We observe that the size of clusters returned by Algorithm 1 increases after each iteration until they cover almost the entire input set. The clustering performance of the model also generally improves with each iteration until it saturates, as we show in Section 5. We also note that cluster assignments become more stable with subsequent iterations, which also leads to decrease in variance across multiple runs. That is, the variance across multiple runs decreases if we run *Kingdra* for more iterations.

## 5 EXPERIMENTS

In this section we evaluate the performance of *Kingdra* on several popular datasets. For a fair comparison, we use the same data pre-processing and same model layer sizes as prior work Hu et al. (2017).

### 5.1 DATASETS

We evaluate *Kingdra* on three image datasets and two text datasets: **MNIST** is a dataset of 70000 handwritten digits of 28-by-28 pixel size. Here, the raw pixel values are normalized to a range 0-1 and flattened to vector of 784 dimensions. **CIFAR10** is a dataset of 32-by-32 color images with 10 classes having 6000 examples each. **STL** is a dataset of 96-by-96 color images with 10 classes having 1300 examples each. For CIFAR10 and STL raw pixels are not suited for our goal as the color information dominates, hence as mentioned in Hu et al. (2017), we use features extracted from a Resnet-50 network pre-trained on the ImageNet dataset. **Reuters** is a dataset containing English news stories with imbalanced data and four categories. We used the same pre-processing as used by Hu et al. (2017); after removing the stop-words, tf-idf features were used. **20News** is a dataset containing newsgroup documents with 20 different newsgroups. Similar to Hu et al. (2017), we remove stop words and keep 2000 most frequent words, and used tf-idf features. All our experiments were performed using the same pre-processed data.

### 5.2 EVALUATION METRIC

We use standard unsupervised evaluation methodology and protocol to compare different methods. Following Xie et al. (2016), we set the number of clusters the same as the number of ground truth

| Method | MNIST | STL | CIFAR10 | Reuters | 20news |
|---|---|---|---|---|---|
| $K$-means | 53.3 (0.1) | 85.0 (0.2) | 34.4 (0.9) | 53.7 (0.4) | 14.0 (1.5) |
| AC | 62.1 (0.0) | 82.2 (0.0) | 42.4 (0.0) | 54.9 (0.0) | 18.6 (0.0) |
| dAE+$K$-means | 67.7 (3.0) | 20.8 (1.9) | 45.2 (2.1) | 33.7 (0.2) | 7.9 (0.1) |
| dVAE+$K$-means | 65.2 (3.4) | 60.8 (1.9) | 44.2 (0.2) | 53.7 (1.4) | 12.2 (0.2) |
| DEC Xie et al. (2016) | 84.3 | 78.1 (0.1) | 46.9 (0.9) | 67.3 (0.2) | 30.8 (1.8) |
| DeepCluster Caron et al. (2018) | 27.9 (1.7) | 69.9 (3.2) | 37.2 (0.5) | 43.1 (4.3) | 15.8 (1.2) |
| Deep RIM Hu et al. (2017) | 58.5 (3.5) | 92.5 (2.2) | 40.3 (3.5) | 62.3 (3.9) | 25.1 (2.8) |
| IMSAT (RPT) Hu et al. (2017) | 89.6 (5.4) | 92.8 (2.5) | 45.5 (2.9) | **71.9 (6.5)** | 24.4 (4.7) |
| IMSAT (VAT) Hu et al. (2017) | 98.4 (0.4) | 94.1 (0.4) | 45.6 (0.8) | 71.0 (4.9) | 31.1 (1.9) |
| LADDER-IM (ours) | 95.0 (2.8) | 90.7 (1.8) | 49.5 (2.9) | 68.2 (2.8) | 38.4 (2.5) |
| LADDER-IM-ensemble (ours) | 95.1 (0.4) | 91.5 (0.3) | 51.5 (0.9) | 69.0 (3.4) | 40.5 (0.6) |
| LADDER-DOT (ours) | 89.2 (7.2) | 76.1 (4.7) | 48.0 (1.0) | 66.6 (4.8) | 25.6 (1.3) |
| KINGDRA-LADDER-DOT (ours) | 98.0 (0.01) | 93.5 (1.4) | 54.3 (2.5) | **71.9 (3.4)** | 28.4 (1.2) |
| KINGDRA-LADDER-IM (ours) | **98.5 (0.4)** | **95.1 (0.1)** | **54.6 (0.9)** | 70.5 (2.0) | **43.9 (1.4)** |

Table 2: Comparison of clustering accuracy (%) on five benchmark datasets. Averages and standard deviations are reported. The results for DEC, Deep RIM, and IMSAT are excerpted from Hu et al. (2017).

classes and evaluated unsupervised clustering accuracy as:

$$\text{ACC} = \max_p \frac{\sum_{i=1}^N \mathbf{1}\{l_n = p(c_i)\}}{N}, \tag{4}$$

where $l_i$ and $c_i$ are the ground truth cluster label and the cluster label assigned by the model respectively. We find the best one-to-one mapping of ground truth label and model generated clusters with $p$ ranging over all one-to-one mappings.

## 5.3 COMPARED METHODS

We compare *Kingdra* against several clustering algorithms on our datasets. Specifically, we compare against traditional clustering algorithms such as **K-Means** and Agglomerative clustering(**AC**). We also compare against representation learning baselines where we use models such as Deep Autoencoders(**dAE**), Deep Variational Auto-encoders (**dVAE**), and then use K-Means on the learned representations. Finally, we also compare our model with deep learning based clustering methods such as **Deep RIM**, **DEC**, **DeepCluster**, and **IMSAT**. Deep RIM uses a multi-layer neural network with the RIM objective. DEC iteratively learns a lower dimensional feature representation and optimizes a clustering objective. We also compare with two versions of IMSAT – **IMSAT(RPT)** and **IMSAT(VAT)** where data augmentation is used to impose invariance in the model outputs. For our results, we report the performance of *Ladder-IM* and *Ladder-Dot* individually, and finally **Kingdra** that includes an ensemble of *Ladder-\** networks, along with the semi-supervised iterations. For a fair comparison we used the same network architecture for all the neural network based models.

## 5.4 EXPERIMENTAL RESULTS

Accuracy results of prior approaches and ours is shown in Table 2. As can be seen from the table, *Ladder-IM* by itself delivers good performance and *Kingdra-Ladder-IM* achieves higher clustering accuracy than state-of-the-art deep unsupervised approaches such as DEC Xie et al. (2016) and IMSAT Hu et al. (2017) in all five data sets. Further, the gap between *Kingdra* and prior approaches is significant in two data sets: *Kingdra-Ladder-IM* achieves an average accuracy of 54.6% for CIFAR10 compared to 45.6% for IMSAT and 46.9% for DEC – an 8% increase in absolute accuracy. Similarly, *Kingdra-Ladder-IM* achieves an average accuracy of 43.9% for 20news compared to 31.1% for IMSAT and 30.8% for DEC – an increase of over 12% in absolute accuracy. Note that while deep networks are state-of-the-art for most data sets, linear approaches outperform deep approaches on 20news with linear RIM achieving 50.9% accuracy Hu et al. (2017). We also tried DeepCluster Caron et al. (2018) in our experimental setting, but observed the model to degenerate, assigning most of the samples to the same cluster. Additional analysis of DeepCluster is in the Appendix.

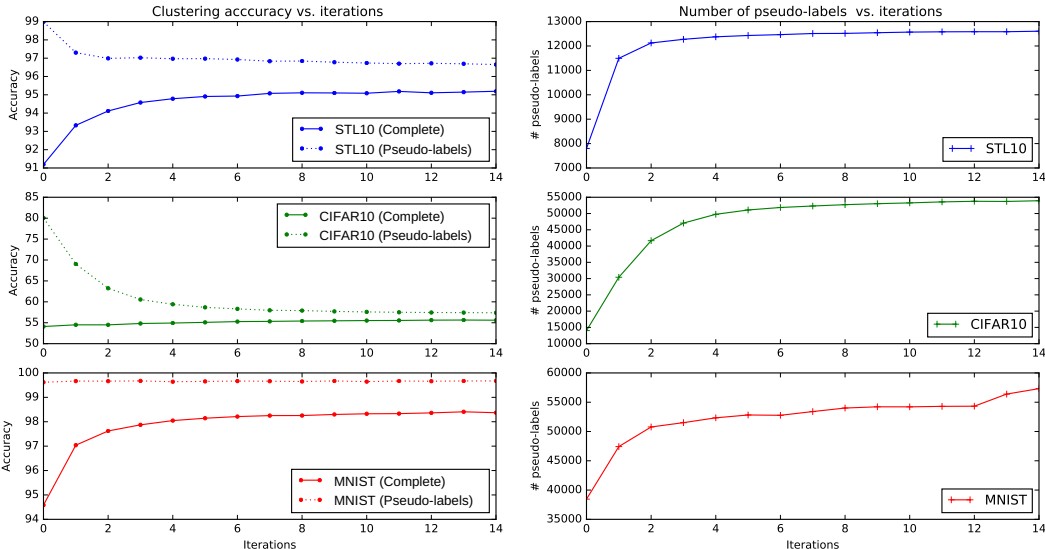

Figure 2: The left graph shows clustering and pseudo-label accuracy vs iterations for STL, CIFAR10, and the MNIST datasets. The right graph shows the number of pseudo-labels vs iterations.

An interesting aspect to note is that the use of an ensemble by itself only provides small gains of 1-2%, similar to what one expects from ensembles in supervised learning (e.g., compare *Ladder-IM* with *Ladder-IM*-ensemble). The large gains mainly come from *Kingdra* using the ensemble to generate pseudo-labels, which is then iterated. For example, *Kingdra-Ladder-IM* provides absolute gains of 4-6% in most data sets over the base model. Similarly, *Kingdra-Ladder-Dot* provides absolute gains of 9% in MNIST and 17% in STL over the base *Ladder-Dot* model. Thus, our approach of generating pseudo-labels from ensembles is a powerful approach that delivers large gains in unsupervised learning.

Also note that *Kingdra-Ladder-IM* performs better than *Kingdra-Ladder-Dot* for most data sets except for the Reuters data set where the latter performs better (Reuters has a large class imbalance with the largest class representing 43% of the data).

Finally, note the standard deviation of the various approaches shown in the Table. One can see that *Kingdra* in general results in lower standard deviation than many of the prior approaches even while delivering higher accuracy.

Figure 2 shows the accuracy of pseudo-labels and *Kingdra-Ladder-IM*, as well as number of pseudo-labels identified by the graph clustering algorithm vs the number of iterations for STL, CIFAR10, and MNIST datasets. As iterations progress, the accuracy of pseudo labels decreases as more pseudo-labels get added; however, this still helps improve the overall clustering accuracy. Note that, unlike pure semi-supervised approaches which use a small set of (randomly sampled) data points that match the input data distribution, our pseudo-labels do not completely match the input data distribution (since our selection algorithm is biased towards easy data points). This causes an increased gap between the accuracy of pseudo-labels, and that of overall clustering.

## 5.5 QUALITATIVE ANALYSIS

Figure 3 shows the similarity graph obtained after the first three iterations of *Kingdra* on the MNIST dataset.As the iteration progresses, one can see that there are fewer inter-cluster linkages indicating that the models are converging on the labels for these data points. Figure 4 shows randomly selected examples from our final clusters generated by *Kingdra*. One can see that the examples are highly accurate for MNIST, thus resulting in high overall accuracy. However, for CIFAR10, there are several incorrectly labelled examples, including two clusters which do not have a clear mapping with any ground truth class, thereby resulting in much lower overall accuracy.

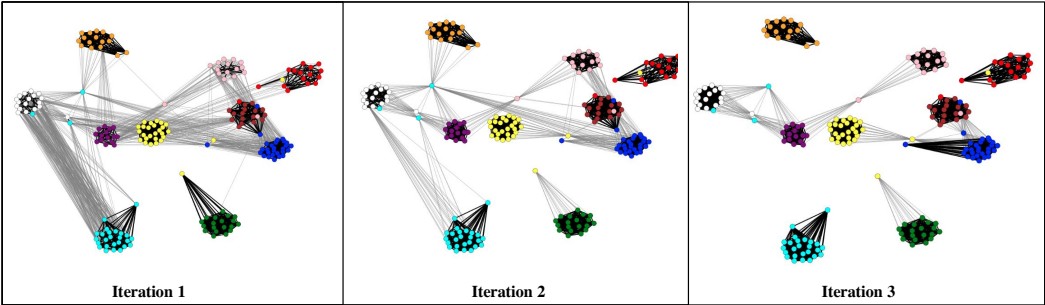

Figure 3: Similarity graph from a sample of the input data points, obtained during three iterations of *Kingdra* for MNIST. The strong positive edges are shown by black lines and grey lines indicate that at least two models think they belong to the same class. The different colors show different true class labels.

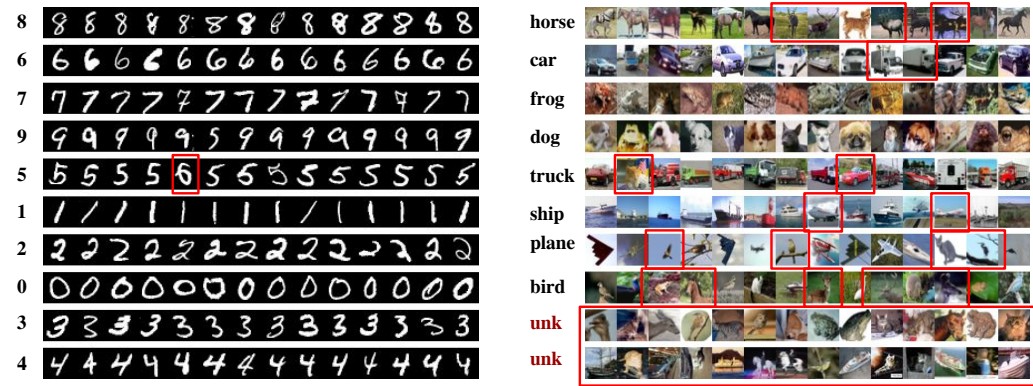

Figure 4: Examples of randomly selected images obtained from our final clusters for MNIST and CIFAR10 datasets. The images with incorrect class associations are identified by red boxes.

## 6 CONCLUSION

In this paper, we introduced *Kingdra*, a novel pseudo-semi-supervised learning approach for clustering. *Kingdra* outperforms current state-of-the-art unsupervised deep learning based approaches, with 8-12% gains in absolute accuracy for CIFAR10 and 20news datasets. As part of *Kingdra*, we proposed clustering ladder networks, *Ladder-IM* and *Ladder-Dot*, that works well in both unsupervised and semi-supervised settings.

## 7 DISCUSSION

While *Kingdra* performs well in the datasets we studied, the similarity-based graph clustering algorithm used has difficulty as the number of classes increase. For example, for the datasets we evaluated, the $t_{pos}$ and $t_{neg}$ can be simply set to the number of models in the ensemble. However, as the number of classes increase, these thresholds may need some tuning. For CIFAR100, with 100 classes, our graph clustering algorithm is not able to identify 100 diverse classes effectively. We are looking at improving the clustering algorithm as part of future work. We are also evaluating adding diversity to the models in the ensemble, either via changing the model structure, size and/or through changing the standard deviation of random noise used in ladder networks.

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

## A   APPENDIX

## B   *Ladder-\**: LADDER NETWORKS FOR CLUSTERING

We now describe the *Ladder-\** architecture for the individual models in the ensemble. We use the same model architecture for both unsupervised learning in the initial step, and the subsequent semi-supervised learning iterations, the only difference being that the semi-supervised models carry an extra supervision loss term. Our architecture augments ladder networks Rasmus et al. (2015) with one of two losses – an information maximization loss similar to the RIM method described in Krause et al. (2010); Hu et al. (2017), or a dot product loss Chang et al. (2017). We call the two variants *Ladder-IM* and *Ladder-Dot*, respectively. We first briefly describe the RIM method and ladder networks, followed by our *Ladder-IM* and *Ladder-Dot* methods.

REGULARIZED INFORMATION MAXIMIZATION (RIM)

The Regularized Information Maximization (RIM) approach for unsupervised learning was introduced in Krause et al. (2010) and extended by Hu et al. (2017) for multi-dimensional setting. The RIM method minimizes the following objective for a classifier:

$$R(\theta) - \lambda I(X; Y) \tag{5}$$

where $R(\theta)$ is a regularization term, and $I(X; Y)$ is the mutual information between the input $X$ and output $Y$ of the classifier. The mutual information can be written as the difference between marginal entropy and conditional entropy Hu et al. (2017):

$$I(X; Y) = H(Y) - H(Y|X) \tag{6}$$

where $H(.)$ and $H(.|.)$ are entropy and conditional entropy, respectively. Maximizing the marginal entropy term $H(Y)$, encourages the network to assign disparate classes to the inputs, and thus encourages a uniform distribution over the output classes. On the other hand, minimizing the conditional entropy encourages unambiguous class assignment for a given input. In the unsupervised setting, where other priors are not known, this loss makes intuitive sense.

For the regularization loss term $R(\theta)$ above, many options have been proposed. Hu et al. (2017), for example, propose a Self-Augmented Training (SAT) loss, which imposes invariance on the outputs of original and slightly perturbed input data. The authors experimented with random perturbation (IMSAT-RPT), and adversarial perturbation (IMSAT-VAT) where the perturbation is chosen to maximize the divergence between the two outputs on the current model.

LADDER NETWORKS

Ladder networks Rasmus et al. (2015) have shown impressive performance for semi-supervised classification. They employ a deep denoising auto encoder architecture, in which an additive noise is added to each hidden layer in the encoder, and the decoder learns a denoising function for each layer. The objective function is a weighted sum of supervised cross entropy loss on the output of the noisy encoder, and a squared error of the unsupervised denoising loss for all layers. Unlike standard auto-encoders, ladder networks also add lateral skip connections from each layer of the noisy encoder to the corresponding decoder layer. The additive noise acts as a regularizer for the supervised loss, while the lateral connections in the denoising decoder layers enable the higher layer features to focus on more abstract and task-specific features. See Pezeshki et al. (2016) for a detailed analysis.

Borrowing the formalism in Pezeshki et al. (2016), a ladder network with $L$ encoder/decoder layers can be defined as:

$$\tilde{x}_i, \tilde{z}_i^{(1)}, ..., \tilde{z}_i^{(L)}, \tilde{y}_i = \text{Encoder}_{noisy}(x_i, \theta_j),$$
$$x, z_i^{(1)}, ..., z_i^{(L)}, y_i = \text{Encoder}_{clean}(x_i, \theta_j),$$
$$\hat{x}_i, \hat{z}_i^{(1)}, ..., \hat{z}_i^{(L)} = \text{Decoder}(\tilde{z}_i^{(1)}, ..., \tilde{z}_i^{(L)}, \phi_j),$$

where $\theta_j$ and $\phi_j$ are the parameters for the Encoder and Decoder, respectively. The variables $z_i^{(k)}$, $\tilde{z}_i^{(k)}$, and $\hat{z}_i^{(k)}$ are the hidden layer outputs for the clean, noisy, and denoised versions at layer $k$, respectively. $x$, $y_i$, $\tilde{y}_i$ are the input, clean output and the noisy output, respectively. The objective function consists of the reconstruction loss between clean and decoded intermediate features:

$$loss^{denoise} = \Sigma_{i=1}^n \Sigma_{k=1}^L \lambda_k^{denoise} \left\| (z_i^{(l)}, \hat{z}_i^{(l)}) \right\|_2 \tag{7}$$

and a supervised cross entropy loss on the output of the noisy encoder (which is used only in the semi-supervised setting):

$$loss^{sup} = -\Sigma_{i=1}^n log P(\tilde{y}(i) = y^*|x(i)) \tag{8}$$

*Ladder-IM & Ladder-Dot*

We now describe our novel *Ladder-IM* and *Ladder-Dot* models. The unsupervised denoising loss in Equation 7, along with the lateral connections architecture enables ladder networks to learn useful features from unsupervised data. However, in the absence of any supervised loss (Equation 8), ladder

networks can degenerate to the trivial solution of a constant output for each encoder layer, as the decoder can then simply memorize these constants to make the denoising loss zero. Having batch normalization layers helps to alleviate this problem, but the loss function still allows the trivial solution. On the other hand, the mutual information loss (Equation 6) in RIM methods, in particular the marginal entropy term $H(Y)$, encourages the network to assign disparate classes to the inputs.

***Ladder-IM***: Combining ladder networks with information maximization can fix the above degeneracy problem, while simultaneously encouraging the ladder output towards a uniform distribution. We use both the clean, and noisy outputs of the ladder network for computing the mutual information loss, i.e.

$$loss^{MI} = I(X; \tilde{Y}) + I(X; Y) \tag{9}$$

where $Y = \{y_1, \ldots, y_N\}$ is the set of clean outputs, and $\tilde{Y} = \{\tilde{y}_1, \ldots, \tilde{y}_N\}$ is the set of noisy outputs from the ladder network.

Another way of thinking about the *Ladder-IM* approach is completely within the RIM framework. The unsupervised ladder loss $loss^{denoise}$, can be simply thought of as the regularization term $R(\theta)$ in equation 5. To that effect, we also add another regularization loss term, which is the KL divergence between the clean and noisy outputs of the ladder network encoder, i.e.

$$loss^{ladder\_R} = KL(p(\tilde{y}|x), p(y|x)) \tag{10}$$

This regularization can be thought of as a generalization of the random perturbation loss proposed in Hu et al. (2017), where the authors impose invariance on the outputs of original and randomly perturbed inputs. Our regularization based on adding noise to the hidden layers is similar to dropout Srivastava et al. (2014), and can be thought of as adding higher level feature noise, rather than just input noise.

Thus, in the unsupervised case, this would lead to the following minimization objective:

$$loss^{Ladder\text{-}IM} = loss^{denoise} + \alpha \cdot loss^{ladder\_R}$$
$$+ \beta \cdot loss^{MI} \tag{11}$$

In this paper, we set $\alpha$ and $\beta$ to one. Finally, in the semi-supervised case, we also add the supervised cross entropy term (Equation 8), as done in the original ladder networks.

***Ladder-Dot***: We also try a dot product loss to fix the above degeneracy problem. The dot product loss is defined to be

$$D(X_i, X_j) = Y_i^T Y_j, \text{ if } i \neq j \tag{12}$$

which forces the network outputs for different inputs to be as orthogonal as possible. This has a similar effect to IM loss, encouraging the network to assign disparate classes to the inputs.

Among *Ladder-IM* and *Ladder-Dot*, we found *Ladder-IM* to perform better than *Ladder-Dot* in most cases. However, we did find that *Ladder-Dot* along with *Kingdra* iterations outperforms when the data set has a large imbalance in the number of samples per class. The reason for this is that the dot product loss is agnostic to the number of samples per class, while the marginal entropy term in the IM loss will drive the network towards overfitting a class with more samples, compared to a class with less number of samples.

Overall, we found in our experiments that *Ladder-IM* showed superior performance to IMSAT-RPT and IMSAT-VATHu et al. (2017) on most data sets. Moreover, in pure semi-supervised settings also, *Ladder-IM* outperformed vanilla ladder networks in our preliminary analysis.

## C    EXPERIMENTAL RESULTS

### C.1    IMPACT OF NUMBER OF MODELS IN ENSEMBLE

We evaluated the accuracy of KINGDRA-LADDER-IM as the number of models in the ensemble was varied. MNIST accuracy with 1, 2, 5, 10, and 15 models is 95.0, 96.2, 97.4, 98.5, and 98.5 respectively. This suggests that accuracy saturates after 10 models and we use 10 models for our ensemble for all our experiments.

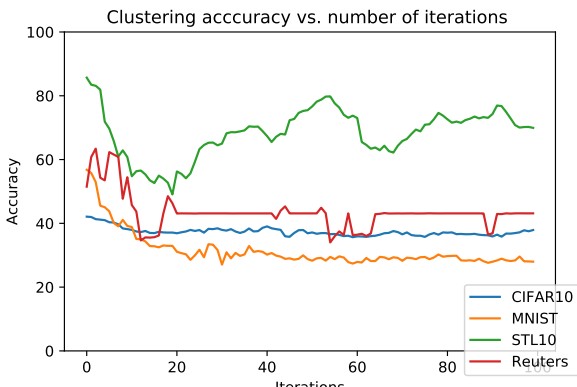

Figure 5: Graph shows clustering accuracy vs iterations for DeepCluster. We see that there is no improvement in accuracy after the first iteration.

## C.2 COMPUTATION COST

We have an efficient implementation of clustering, which takes 210s for largest n = 70000. On a server with four P100 GPUs, CLadder-IM takes 2mins, CLadder-IM with ensemble takes 8mins and Kingdra with 10 iterations takes 80mins while IMSAT(RPT) takes 5mins.

## C.3 ANALYSIS OF DEEPCLUSTER

Here we give an analysis of DeepCluster Caron et al. (2018), explaining the shortcomings. We observed that the clustering accuracy generally decreases with iterations. This is because the pseudo-labels generated could be bad, which results in worse accuracy in the next iteration. On the other hand, our approach only uses small number high-confidence samples for pseudo-labels.

## D DETAILS OF THE DATASETS

- **MNIST**: A dataset of 70000 handwritten digits of 28-by-28 pixel size. The raw pixel values are normalized to a range 0-1 and flattened to vector of 784 dimensions.
- **CIFAR10**: A dataset of 32-by-32 color images with 10 classes having 6000 examples each. Similar toHu et al. (2017), features are extracted using 50-layer pre-trained deep residual networks.
- **STL**: A dataset of 96-by-96 color images with 10 classes having 1300 examples each. We do not use the 100000 unlabeled images provided in the dataset. Similar to Hu et al. (2017)], features are extracted using 50-layer pre-trained deep residual networks.
- **Reuters**: A dataset containing English news stories with four categories : corporate/industrial, government/social, markets, and economics. We used the same pre-processing as used by Hu et al. (2017). After removing the stop-words, td-idf features were used.
- **20News**: A dataset containing newsgroup documents with 20 different newsgroups. Similar to Hu et al. (2017) after removing stop words and keeping 2000 most frequent words, td-idf features were used.

