# OpenReview forum: "Unsupervised Clustering using Pseudo-semi-supervised Learning"
_ICLR.cc/2020/Conference — Accept (Poster)_

### Official Review · AnonReviewer3 · 2019-10-18
**Official Blind Review #3**

**Rating:** 6

**Review:**

This paper proposes a method for unsupervised clustering. Similarly to others unsupervised learning (UL) papers like "Deep Clustering for Unsupervised Learning of Visual Features" by Caron et al., they propose an algorithm alternating between a labelling phase and a training phase. Though, it has interesting differences. For example, unlike the Caron et al. paper, not all the samples get assigned a labels but only the most confident ones. These samples are determined by the pruning of a graph whose edges are determined by the votes of an ensemble of clustering models. Then, these pseudo labels are used within a supervised loss which act as a regularizer for the retraining of the clustering models.

Novelties /contributions/good points:
* Votes from the clustering models to create a graph
* Using a graph to identify the most important samples for pseudo labelling
* Modification of the ladder network to be used as clustering algorithm
* Good amount of experiments and good results

Weaknesses:
* The whole experiment leading to Table 1 in page 2 is unclear for me. I have trouble understanding the experiment settings. Could you please rephrase it. About initial/ final clustering for example and the rest as well. The whole thing puzzles me whereas the experiments section at the end is much more clear.
* Lack of motivation about why using the Ladder method rather than another one. Other recent methods have better results in semi-supervised learning.
* Algorithm 1 seems quite ad-hoc. Do more principled algos exist to solve this problem ? You could write about it and at least explain why it would not be feasible here. The sentence "The intuition is that most of the neighbours of that node will also be connected with each other" is unmotivated: no empirical proof for this ?
* Related work section is too light. It is an important section and should really not be hidden or neglected.
* In the experiments, you could add the "Deep Clustering for Unsupervised Learning of Visual Features"  as baseline as well even if they use it for unsupervised learning as they do clustering as well.
* In the experiments, you use the features extracted from ResNet-50 but what about finetuning this network rather than adding something on top or even better starting from scratch. Because here CIFAR-10 benefits greatly from the ImageNet features. I know that you should reproduce the settings from other papers but it might be good to go a bit beyond. Especially, if the settings of previous papers are a bit faulty.
* Regarding, the impact of number of models in section D of the appendix, there is no saturation at 10 models. So how many models are necessary for saturation of the performance ?
* Minor point: several times, you write "psuedo".

Conclusion: the algorithm is novel and represents a nice contribution. Though, there are a lot of weaknesses that could be solved. So, I am putting "Weak accept" for the moment but it could change towards a negative rating depending on the rebuttal.


**Experience Assessment:**

I have published one or two papers in this area.

**Review Assessment: Checking Correctness Of Derivations And Theory:**

I carefully checked the derivations and theory.

**Review Assessment: Checking Correctness Of Experiments:**

I carefully checked the experiments.

**Review Assessment: Thoroughness In Paper Reading:**

I read the paper thoroughly.

---

> ### Author Response · Authors · 2019-11-12
> **Response to reviewer #3**
>
> Thank you for your comments and the positive review of the paper.
>
> * We have updated the paper to clarify these experiments better. At a high level, the goal of the experiments in page 2 is to see the effect of generating pseudo labels using existing approaches on the final clustering accuracy using our iteration-based approach. These experiments establish two things: 1) We need good quality of initial pseudo labels to get good final clustering accuracy. 2) None of the existing methods provide high accuracy pseudo labels.
>
> * Yes, there are several recent methods for semi-supervised learning that have higher accuracy than ladder networks.  For some of these approaches [1,2],  data-augmentation is a core component which assumes some domain knowledge of the dataset. Further, many of the data-augmentation techniques are specific to image datasets. There are other  methods [3,4] which uses adversarial training to learn latent features. However, we found that these methods do not work well if we jointly train them with unsupervised losses. Ladder networks does not require any domain-dependent augmentation, works for both image and text datasets, and can be easily jointly trained with supervised and unsupervised losses. Thus, we chose to work with Ladder networks though our approach is general enough to work with any semi-supervised method that accommodates training with unsupervised loss terms.
>
> * Traditional clustering algorithms focus mainly on clustering the entire data set, not on finding high accuracy clusters of subsets of the data, and thus do not achieve high enough accuracy required for improving final clustering accuracy. One principled algorithm is Girvan–Newman algorithm [5]  that was proposed for community detection but we found that it was computationally impractical given the size of our datasets.
> Regarding the intuition that most of the neighbours of that node will be connected with each other, we found this to be empirically true in our experiments. For example, on Cifar10, for the threshold of 90% models agreeing on the label, about 81% of the nodes in a cluster were connected to each other. If the threshold is at 100%, all nodes in a cluster are connected with each other due to transitivity. We have updated the paper with these numbers.
>
> *  We have updated the related work section with discussion of several other related papers.
>
> * We found that running K means starting with a random initialization to assign pseudo-labels as described in the paper resulted in poor pseudo-label accuracy. Further, if we iterate based on these low accuracy pseudo-labels, the model degenerates to assigning most of the samples to the same cluster. Thus, we felt that it was unfair to the authors to add these results as a baseline, especially since the authors themselves did not report clustering performance. Note that, for the results in section 2, we did not start with a random initialization (we used a ladder network trained with an unsupervised loss to generate the initial pseudo-labels).
>
> * We did try some experiments on not using any pre-trained models for features and training convnets from scratch. On the cifar10 dataset, using Resnet34 as CNN initialized randomly, our method was able to achieve clustering accuracy of 35.17 ( achieving about 2% improvement over the same model without our framework) . In the literature, there are a couple of papers [6 , 7 ] that performs clustering on cifar-10 datasets from scratch, but they use a variety of domain-based data augmentation-based techniques to improve performance and we were not able to reproduce their results. Furthermore, they are applicable to only image datasets and do not help with text-based datasets that we also evaluate on.
>
> * We ran additional experiments with 15 models in the ensemble and the accuracy remained at 98.5% accuracy on the MNIST dataset. This suggests that accuracy saturates after 10 models. We have updated the paper with this result.
>
> * Thanks for pointing it out, we have fixed it in the revised version of the paper.
>
> [1] David Berthelot, Nicholas Carlini, Ian Goodfellow, Nicolas Papernot, Avital Oliver, and Colin Raffel.  Mixmatch: A holistic approach to semi-supervised learning
>
> [2]  Qizhe Xie, Zihang Dai, Eduard Hovy, Minh-Thang Luong, and Quoc V Le.  Unsupervised data augmentation
>
> [3] Takeru Miyato, Shin-ichi Maeda, Masanori Koyama, and Shin Ishii. Virtual adversarial training: a regularization method for supervised and semi-supervised learning
>
> [4] Saki Shinoda, Daniel E Worrall, and Gabriel J Brostow.  Virtual adversarial ladder networks for semi-supervised learning
>
> [5] Girvan M. and Newman M. E. J., Community structure in social and biological networks
>
> [6] Jianlong Chang, Lingfeng Wang, Gaofeng Meng, Shiming Xiang, and Chunhong Pan. Deep adaptive image clustering
>
> [7] Xu Ji, João F Henriques, and Andrea Vedaldi. Invariant information clustering for unsupervised image classification and segmentation

---

> > ### Comment · AnonReviewer3 · 2019-11-12
> > **Response to authors**
> >
> > I thank the authors for their detailed answer.
> >
> > 1) Regarding "We did try some experiments on not using any pre-trained models for features and training convnets from scratch.", between training from scratch and using a fully pretrained model, there is a middle point. For example, you could use for a network pretrained with self-supervision as done for semi-supervised learning in "Semi-Supervised Learning with Scarce Annotations" by Rebuffi et al. or "S4L: Self-Supervised Semi-Supervised Learning" by Zhai et al. That could make a stronger case than using a fully pretrained net and better results than from scratch.
> >
> > 2) The explanation for the choice of Ladder networks satisfies me as well as the details for Algo 1.
> >
> > 3) Thanks for the saturation analysis on MNIST.
> >
> > 4)  I would still be interested by a baseline using "Deep Clustering for Unsupervised Learning of Visual Features".

---

> > > ### Author Response · Authors · 2019-11-15
> > > **Response**
> > >
> > > 1) Regarding your comment about using self-supervised networks as a baseline rather than starting from scratch or using pre-trained features. Yes, we agree that it would be an interesting middle point to evaluate. We would like to run this experiment but unfortunately we won’t have enough time to do it before the author response deadline.  Also, for apples to apples comparison, we will need to evaluate prior work under this setting as well.
> > >
> > > 2) We have added DeepCluster as another baseline in Table 2 and added some analysis of DeepCluster in the appendix.

---

### Official Review · AnonReviewer2 · 2019-10-22
**Official Blind Review #2**

**Rating:** 6

**Review:**

This paper proposed an unsupervised learning method of clustering using semi-supervised clustering as a bridge. The method first trains an ensemble of clustering models and use the edge-level majority vote to determine a graph, and then applies rule to get partial clustering signals to feed the final semi-supervised clustering. The scheme is in an iterative fashion to further enhance the quality. I find this paper interesting and somewhat novel, with the following comments.

1. In algorithm 1, is it possible that too many nodes are removed so one cannot get k clusters in the end? Though finding cliques are time consuming, have the authors conducted experiments to see the difference between the real clique finding algorithm and the greedy one proposed?

2. Does the ensemble clustering step have stability issue regarding the method used? If a different clustering method is used, will the graph constructed later change drastically?

3. The writing. First line of section 3, figure 4 seems to point to figure 1. Section 2 seems to have format issue at the beginning. Section 5 could be merged with section 2.

**Experience Assessment:**

I have read many papers in this area.

**Review Assessment: Checking Correctness Of Derivations And Theory:**

I assessed the sensibility of the derivations and theory.

**Review Assessment: Checking Correctness Of Experiments:**

I assessed the sensibility of the experiments.

**Review Assessment: Thoroughness In Paper Reading:**

I read the paper at least twice and used my best judgement in assessing the paper.

---

> ### Author Response · Authors · 2019-11-12
> **Response to reviewer #2**
>
> Thank you for your comments and the positive review of the paper.
>
> 1.  While we did not find identifying clusters to be an issue for the five datasets in our evaluation, identifying k clusters when k is large is indeed challenging (as discussed in Appendix E, applying Algorithm 1 on the Cifar-100 dataset results in fewer than 100 clusters).
>
> Given that the number of nodes in the graph is large (50K-70K), finding cliques, even using approximate algorithms, is prohibitively time consuming. For example, Girvan–Newman algorithm [5] is O( |E|^2 * |V| ).
>
> 2. We performed experiments with two different initial clustering methods (using mutual information loss and using dot product loss, respectively, as unsupervised loss terms, and as described in the paper) . The initial graphs constructed using the two methods were indeed different. Still, we observed improvement in accuracy over iterations using ensemble clustering and the scheme converged empirically. The key requirement for the iteration to work is the presence of some diversity in the graphs extracted from the various models of the ensemble.
>
> 3. Thanks for the suggestion. We have fixed the citation and format issues. For now, we have left section 2 and section 5 as separate since we feel section 2 serves as motivation for some of the decisions we make in the design of our algorithm.
>
> Please let us know if you have any further questions.

---

### Official Review · AnonReviewer1 · 2019-10-26
**Official Blind Review #1**

**Rating:** 6

**Review:**

This paper presents a method where they 1) use an ensemble of networks to cluster unlabeled data and assign pairs of data points a cluster label only if all networks agree that the pair belongs to a cluster 2) use the labeled pairs to create a similarity matrix and find a "tight" cluster or set of points that are all very similar to each other. The paper then uses the "labelled" points for semi-supervised learning with a proposed ensemble of models.

The paper's method of creating high precision labels using their multi-step clustering algorithm with information measures is quite interesting. The experiment results look promising.

**Experience Assessment:**

I do not know much about this area.

**Review Assessment: Checking Correctness Of Derivations And Theory:**

I assessed the sensibility of the derivations and theory.

**Review Assessment: Checking Correctness Of Experiments:**

I assessed the sensibility of the experiments.

**Review Assessment: Thoroughness In Paper Reading:**

I read the paper at least twice and used my best judgement in assessing the paper.

---

> ### Author Response · Authors · 2019-11-12
> **Response to reviewer #1**
>
> Thank you for your comments and the positive review of the paper.

---

### Decision · Program_Chairs · 2019-12-19

**Decision:**

Accept (Poster)

**Comment:**

The authors addressed the issues raised by the reviewers, so I suggest the acceptance of this paper.